# Position: Adopt Machine-Human Collaboration Peer-Review through Computational Research Assessment

## Abstract

Scientific output is outgrowing human review capacity, while AI is already used to draft papers. Authors scale with machines; reviewers largely do not. This asymmetry turns quality control into a bottleneck and increases the risk of both false rejection of high-novelty work and acceptance of flawed results. We propose Computational Research Assessment (CRA) as a discipline-level, method-agnostic agenda for machine-human collaboration in peer review. CRA rests on three principles: treat disagreement as a signal that triggers escalation instead of averaging; make every critique evidence-linked, reproducible, and contestable; and build a community immune system with open corpora, benchmarks, and red-team tests to surface gaming and bias. We map these principles to a co-review engine, a community commons, and theoretical foundations, and we outline near-term pilots and falsifiable commitments, informed by an emerging production-grade pre-review system deployed in the wild.

## 1. Position and Thesis

Scientific knowledge is growing at an unprecedented rate, especially in artificial intelligence (AI). Top conferences are deluged with submissions; NeurIPS 2025 received 21,575 valid paper submissions, a roughly 61% increase over 2024 (Comm. Chairs, 2025). This growth far outpaces human review capacity. Meanwhile, researchers increasingly leverage large language models (LLMs) to generate research text and even entire drafts (Naddaf, 2025b). By 2024, one in seven biomedical papers showed linguistic signatures of AI writing (Mallapaty, 2025). More broadly, AI tools are reshaping scientific publishing workflows end-to-end, not just writing, motivating discipline-level assessment agendas (Wang et al., 2023; Conroy, 2023; Charness et al., 2025; Luo et al., 2025). A recent analysis (Naddaf, 2026) suggests that many reviewers are already using AI tools for peer review, often against official guidance. If peer review remains a purely human endeavor, it will crack under this pressure/reality (Wei et al., 2025; Kim et al., 2025). We risk missing breakthrough ideas or letting flawed work slip through because overwhelmed reviewers cannot cope.

Our position is straightforward: averaging review scores is no longer defensible in the age of abundant research and assistive models. The system should respond to variance, not only to central tendency. When reviewers and assessors disagree, the process should slow down in a focused way, directing extra attention to cases that show both strong promise and serious concern. This protects high novelty work and reduces wrongful rejections.

We advance a discipline-level program: computational research assessment (CRA), an emerging paradigm in which algorithmic systems systematically evaluate research articles to assist (not replace) human judgment. CRA is not about ranking scientists or counting citations; it focuses on papers themselves, analyzing text, data, figures, and claims to provide structured insights on quality, rigor, novelty, and potential impact. In essence, CRA off-loads drudgery and consistency-checking to AI while amplifying human insight when it matters most.

A recurring misconception is to equate CRA with the current wave of LLM-assisted reviewing tools. We stress that CRA is a discipline-level agenda to formalise and systematise machine-augmented evaluation, regardless of the specific AI architecture in vogue. Just as artificial general intelligence (AGI) research will persist even if the LLM paradigm is eventually superseded, CRA must remain method-agnostic: LLMs are today's workhorse, but future assessors may be multi-modal foundation models, symbolic–neuro hybrids, or entirely new cognitive engines. Conflating CRA with "LLM peer review" therefore risks short-changing both its longevity and its scientific breadth.

In this position paper, we outline a unified CRA framework, including guiding principles, key metrics, and safeguards against gaming. We draw on existing literature and technologies for AI-assisted peer review, as well as perspectives from the *science of science* and *philosophy of science*, to support our position. Ultimately, we argue that *CRA should emerge as an independent discipline* with its own theoretical

Preliminary work. Under review by the International Conference on Machine Learning (ICML). Do not distribute.

foundations to ensure the integrity and efficiency of research evaluation in the AI era. Section 4 engages credible objections to CRA and clarifies scope, governance, and falsifiable commitments.

CRA must be robust to adversaries. Prompt-injection strings embedded in manuscripts (e.g., invisible white-on-white text) can coerce an AI reviewer toward uniformly positive verdicts. Even if this evades standard categories of fabrication, falsification, or plagiarism, it is unethical manipulation. CRA's "immune-system" pillar should make such attacks detectable and technically defensible.

## 2. The Subjectivity of Scientific Value and the Need for Diverse Evaluation

Judging the value of scientific research is not a straightforward objective task – it is inherently *subjective and nuanced*. Seasoned researchers often speak of having a certain "taste" in research problems, an eye for what is promising or profound. Like tastes in art or cuisine, these judgments vary widely: what one expert hails as groundbreaking, another might dismiss as trivial or misguided. This subjectivity is not a flaw of science, but a reflection of its creative nature. However, it poses a challenge for any assessment system, human or AI: *how do we ensure truly novel, path-breaking ideas get recognized and supported, rather than drowned out by majority opinion or procedural uniformity?*

The traditional peer review system already grapples with this. Studies have shown that peer review outcomes are highly inconsistent – almost *random* in many cases. For instance, a classic experiment resubmitted already-published papers to journals and found that 8 out of 9 were rejected the second time, with reviewers citing "serious flaws" in work that had earlier been deemed worthy (Cortes & Lawrence, 2021). And a controlled test at NeurIPS 2014 famously revealed that whether a given paper was accepted was a 50-50 proposition: "one-half to two-thirds" of accepted papers would have been rejected under an independent review committee (Langford & Guzdial, 2015). These findings underscore that peer review is far from an exact science: it depends greatly on *who* the reviewers are, and their individual biases and perspectives.

One reason for such variance is precisely that *truly innovative work polarizes opinion*. A bold new idea may have obvious weaknesses or depart from prevailing theories, causing many to balk; yet a few reviewers with the right insight will recognize its potential. In the history of science, many revolutionary ideas (from the theory of plate tectonics to the discovery of quasicrystals) faced skepticism or hostility before ultimately winning the day. Often it took *one champion* to push the idea through initial resistance. If instead we had averaged the scores of all reviewers, these ideas might have

been *filtered out as too controversial or divisive*.

This is a critical lesson as we consider AI's role in peer review. An overly mechanistic AI system that seeks consensus or uniformity could inadvertently reinforce *conservative tendencies*, rewarding safe, incremental work that everyone agrees is "pretty good", while penalizing the rare paper that splits opinions sharply. A useful analogy is that if we treat each review like a vote and simply average them, we can blur distinct signals into an uninformative middle. *We must avoid the trap of converging to mediocrity*. The aim of peer review, and by extension CRA, should not be to select the least objectionable papers, but to identify outstanding contributions even when they are polarizing and imperfect.

## 3. Computational Research Assessment

Given the challenges and principles outlined above, we now articulate our vision for CRA as a unified framework. CRA is not merely a grab-bag of AI tools bolted onto the existing process; we propose it as a coherent discipline with its own theoretical underpinnings and a well-defined pipeline. The ultimate aim is to make research evaluation more systematic, scalable, and insightful *without sacrificing fairness or creativity*. Here we describe the core components and workflow of CRA, as well as the foundational principles that would guide its development.

There are three pillars that constitute CRA: (i) a *human-AI co-review engine*, which performs venue-aware pre-review, produces evidence-linked assessments along the N-R-I-O-E (Novelty, Rigor, Impact, Openness, Ethics) dimensions, and supports interactive escalation when human and model judgments diverge; (ii) a *community commons*, which aggregates anonymized manuscripts, reviews, and outcomes to power open benchmarks, policy deliberation, and red-teaming – serving as both knowledge hub and immune system; and (iii) *theoretical foundations*, which formalize constructs such as novelty and rigor, provide explainability and fairness guarantees, and develop mechanism-design defenses against gaming. Taken together, these pillars off-load routine checks to machines while amplifying human judgment, ensure continuous learning under community oversight, and ground CRA in principles that remain method-agnostic as underlying AI technologies evolve.

### 3.1. Human-AI Co-Review Engine

At the heart of CRA is a re-imagined review process that tightly couples AI analysis with human expertise. We propose a two-stage *human-AI co-review engine*, with a critical mechanism to handle disagreement between AI and human.

- *Stage A: AI-Powered Pre-Review*. When a paper is submitted, it first undergoes an automated "rapid review" by specialized AI agents configured for the venue and field.

*Table 1.* Minimal evidence and decision cues for N-R-I-O-E.

| Dimension | Evidence and cue |
| --- | --- |
| Novelty | Unusual combination of concepts with citations to nearest prior art. Short note on distance in a citation graph or an embedding space. Link to examples with similar structure. |
| Rigor | Explicit chain of inference from data to claim. Pointers to tests, confidence intervals, or proofs. Result of simple statistical sanity checks. |
| Impact | A short statement of possible downstream use, required conditions, and replication risk. |
| Openness | Link to a minimal script that runs. Hash of the artifact. Short output log. |
| Ethics | Data license note. Consent or exemption. Risk register entry with severity and mitigation. |

These agents parse the manuscript's content (text, figures, references, perhaps even code/data supplements) and evaluate it along key dimensions. We propose an *N-R-I-O-E* schema as a guiding set of evaluation metrics, which can be further refined. For each dimension, the AI provides a score or qualitative assessment supported by *explicit evidence*. Table 1 summarizes the minimal evidence and decision cues associated with each dimension. The AI then produces a structured review draft whose N-R-I-O-E claims are explicitly evidence-linked (as required by Table 1), including nearest-prior-art pointers for novelty, localized rigor flags tied to specific claims/sections, artifact/openness checks, and ethics risk notes, aiming for grounded novelty judgment as in (Afzal et al., 2025). Importantly, the AI's tone here is assistive: it identifies issues and strengths but does *not* make a final accept/reject recommendation on its own.

- *Stage B: Interactive Human-AI Review*. Human reviewers (selected experts in the field) then enter the loop, armed with the AI's preliminary report. Rather than starting from scratch, they begin with a structured analysis to respond to, verify, or modify. In this stage, reviewers engage in a *dialogue with the AI's assessment*. For example, a reviewer might see the AI flagged a methodological flaw; if the reviewer disagrees, they can override it but should explain why (such as "The AI is mistaken; the assumption it flagged is actually justified by Smith et al. 2021"). Conversely, if the AI missed something, say a subtle logical gap, the reviewer can add that to the report. The AI can even be prompted interactively: "Check if there are any papers that attempted a similar approach in a different domain," or "Re-evaluate the novelty considering these additional references." This stage is about *amplifying human insight*: the drudge work is largely done, freeing the human to focus on higher-order critique and creative evaluation. The end product of Stage B is a *hybrid review* written by a human reviewer, but richly informed by structured reasoning (Dycke et al., 2025), cross-checks, and suggestions.

*Disagreement Escalation* is a critical mechanism here. If the AI and human reviewer strongly diverge (for instance, the AI gives a paper a glowing appraisal but the human is unimpressed, or vice versa), this triggers a flag in the system. Instead of quietly averaging out, the system would notify the editor (or area chair in a conference) that "Reviewer and AI are in disagreement on key aspects." This could prompt a second human opinion or a moderated discussion. The goal is to ensure that when AI and human perceptions conflict greatly, the case gets extra attention rather than slipping through with a middle-ranged score. Such escalation harnesses the diversity of perspective – it's an opportunity to uncover why the disagreement exists. Perhaps the AI saw promise that the human missed, or the AI was fooled by something the human caught; either way, resolving that yields insight.

Notably, our co-review engine *retains a human-in-the-loop for the final judgement*. Even as AI takes on a large portion of analysis, we envision the ultimate accept/reject decision (or scoring in a review exercise) is made by human editors or program chairs, informed by the augmented reviews. In other words, AI helps to present the evidence and even preliminary evaluations, but *accountability remains with humans* – a principle aligning with calls for AI transparency and responsibility in high-stakes decisions (Zou, 2024). The AI's role is advisor and assistant, not judge and jury.

### 3.2. Community Commons and Continuous Learning

Deploying AI in peer review raises concerns beyond the individual paper: how to ensure the system stays fair, robust, and up-to-date? We propose a *Community Commons* as the second pillar of CRA – a shared hub of knowledge, data, and policy that underpins the AI review systems. This Commons would serve multiple functions:

- *Knowledge Base and Training Data*: It aggregates a large repository of past papers, reviews, and editorial decisions, suitably anonymized and curated, to train and refine AI models. Essentially, it's a constantly growing dataset of "review cases" such as (Zhang et al., 2025; Demetrio et al., 2025). By pooling data from many conferences and journals (with permission and privacy safeguards), the community can train powerful models that *learn from the collective experience* of peer review. For example, models can be trained to recognize the kinds of flaws that typically lead to rejection, or the traits that tend to characterize highly novel papers. Prior work has already begun compiling such datasets. One example is the PeerRead dataset of conference submissions and reviews (Kang et al., 2018), which shows that machine learning (ML) can predict aspects of review scores from the paper text (Li et al., 2020). The Commons would greatly expand this, creating a rich foundation for CRA algorithms.

- *Benchmarking and Consensus Building*: The Commons would act as an *open forum to evaluate CRA tools*. It could host challenge datasets, akin to how ImageNet and GLUE accelerated progress in computer vision and natural language processing (Guo et al., 2023), for tasks such as predicting acceptance decisions or assessing novelty using standardized paper sets with known outcomes or expert ratings. Researchers from both AI and metascience fields could contribute, creating a competitive but collaborative environment to improve CRA methods. Alongside, the community can develop *best practices and guidelines*, essentially the evolving "laws of CRA", through open debate. For instance, what evidence is acceptable for an AI to claim a paper lacks rigor? How to quantify novelty in a way that correlates with later impact? These are not easy questions, and answering them requires input from diverse stakeholders: scientists, ethicists, sociologists of science, and AI experts. The Commons provides the platform for hashing out such principles transparently, rather than leaving it to a few developers behind closed doors.

- *Immune System for Gaming and Bias*. Perhaps most critically, the Community Commons would function as an *immune system* against inevitable attempts to game or corrupt algorithmic assessment. Any evaluation metric or algorithm, once known, can become a target for manipulation – a manifestation of Goodhart's Law ("when a measure becomes a target, it ceases to be a good measure"). For example, when automated plagiarism checks became standard, some unethical authors evaded detection via paraphrasing or synonym replacement. Similarly, if an AI reviewer values particular buzzwords or stylistic cues, authors (or their AI) may optimize for those signals *without* improving substance. The Commons would counter this by (i) *crowdsourcing red-teaming*, where the community probes CRA systems for exploits (e.g., nonsense papers that nonetheless score highly, or hidden white-text prompt injections designed to manipulate the assessor), and shares these attack cases to harden defenses; and (ii) enabling communal oversight to detect and address bias patterns in aggregated data (Thelwall & Kousha, 2023). This cooperative monitoring is crucial for *transparency and fairness*: the Commons becomes a watchdog to "evaluate the evaluators" (Gilestro, 2025), ensuring CRA tools serve science rather than distort it.

In essence, the Community Commons is about making CRA a collective endeavor of the scholarly community, not just a tech product. It acknowledges that peer review is deeply linked to academic culture and values, which must guide the design of any computational system that intervenes in it.

### 3.3. Theoretical Foundations of CRA

For CRA to mature into a standalone discipline, it needs more than ad-hoc tools – it requires *theoretical foundations* that can generalize and guide future developments. We foresee a blend of computational theory, information science, and even philosophy of science coming together to provide this foundation. Here are some key elements we envision:

- *Formal Definitions of Core Constructs*: Concepts like "novelty", "rigor", or "impact" need to be defined in ways that are at least partly measurable. This doesn't mean stripping them to narrow metrics, but rather finding representations that capture their essence. For instance, *novelty* might be modeled in information-theoretic terms as the information gain a paper provides over the existing knowledge distribution. There has been intriguing research quantifying novelty as the *atypical combination of prior knowledge* – e.g., using citation networks (Zhang et al., 2026) or semantic embeddings to detect when a paper bridges disparate domains in a rare way (Fontana et al., 2020; Mukherjee et al., 2017). We can build on such ideas to create a formal novelty score that correlates with expert judgment. *Rigor* could be partially quantified by the completeness and correctness of the methodology; one could develop a logic-based framework to verify whether conclusions follow from data, similar to a proof assistant for experimental design, building on recent work that systematically identifies and localizes methodological errors in papers (Bianchi et al., 2025; Xi et al., 2025). *Impact* (prospective) might draw on network diffusion models or historical patterns of how ideas propagate. The key is to give these terms algorithmic counterparts, even if imperfect, that can be continuously refined. By doing so, CRA moves beyond black-box ML and toward *explainable metrics*: we can say *why* a paper scored high on novelty (e.g., it uses method A on problem B which has never been paired before), rather than just getting a mysterious number.

- *Utility and Game Theory*: We should also frame CRA in terms of incentives and game theory to ensure *immunity to gaming* is baked in. This might involve developing *mechanism design* principles for research assessment. For example, one theoretical goal could be: design a scoring system for papers such that the best strategy for authors to maximize their score is simply to do good science (rather than to engage in metric hacking). This is analogous to designing economic mechanisms where truth-telling is the best strategy. It's a lofty goal, but even partial progress (like identifying particularly gameable aspects and finding alternative measures) will help. The Commons provides empirical data on gaming attempts, but theory can provide guarantees or at least frameworks for thinking about them.

- *Fairness, Bias, and Transparency*: Building on work in AI ethics, CRA theory must address how to measure and mitigate biases in algorithmic evaluation. For example, fairness criteria could be developed so that a CRA system's recommendations do not inadvertently favor one

discipline or demographic. One might require that a paper's content is judged independently of irrelevant attributes, meaning our algorithms should ignore author identities or affiliations by design, effectively enforcing a double-blind principle (Lin et al., 2023). Transparency is another cornerstone: theoretical work on explainable AI can inform how an AI reviewer's "thought process" (e.g., why it flagged a section as problematic) can be communicated in a human-readable form. Perhaps each AI comment in the review comes with a traceable link to evidence (as we have illustrated with citations in this paper). This way, authors and reviewers can *inspect and challenge* the AI's reasoning. An explainable CRA would allow an author to say, "The AI claimed our method failed a statistical test; here is our rebuttal with additional data." By formalizing explainability, one needs to ensure CRA systems are not black boxes imposing opaque judgments on researchers. Instead, it ought to be discussion partners that can be reasoned with.

- *Integrative Models of Peer Review Outcomes*: Finally, a theoretical framework could tie everything together in a model of how decisions are made. For example, one could extend Bayesian decision theory to peer review: consider the "true quality" of a paper as a latent variable, and each reviewer (human or AI) provides a noisy signal about it. How should these signals be combined or weighted to maximize the chance of accepting truly good papers and rejecting unqualified ones? Traditional peer review, implicitly or explicitly, often uses averaging. But theory might show that in high-variance situations, a different aggregation (or no aggregation, just escalation to discussion) is optimal. Some initial models along these lines have been explored in meta-research. CRA can elevate this to a rigorous study: we can A/B-test peer review with and without AI assistance to see how the distribution of outcomes changes, and then optimize the system for better outcomes (e.g., more truly novel papers accepted, fewer required retractions later, etc.).

Developing these foundations is admittedly challenging – peer review has defied simple quantification for decades. However, the very act of trying to do it could yield deep insights. Even if we conclude that some aspects of the judgment cannot be easily formalized, knowing that boundary is valuable. The positive outlook we present is that, with the vast data now available and modern analytical tools, we can finally start to *treat peer review itself as a subject of computational and theoretical study* rather than just an art. This mirrors what happened in other domains: for instance, the once purely intuitive field of algorithm design was put on formal footing by computational complexity theory. We believe a similarly rigorous framework can emerge for research assessment, one that will strengthen trust in any AI-assisted processes.

## 4. Alternative Views

"*Use AI for verification only; and not more than this.*" This view treats verification bandwidth as the main constraint and argues that AI should focus on bounded, auditable verification actions (claim-evidence mapping, artifact reruns, statistical/provenance checks), while avoiding acceptance prediction or scalar scoring (Shah, 2024; You et al., 2026). *Our response:* we acknowledge verification-first stance is important and reliable, but argue CRA must also specify how verified evidence is structured, compared, and routed (e.g., N-R-I-O-E and disagreement-triggered escalation).

"*Keep AI out of peer review entirely.*" This opposing view focuses on operational safety: confidentiality and IP risks, bias and drift in models, and automation bias can shift accountability from expert reviewers to opaque tooling. Under this stance, the right policy is abstinence: keep peer review human and limit AI to author-side writing/support. *Our response:* we treat these concerns as governance requirements, not afterthoughts: any CRA deployment must support strict data controls (e.g., on-prem/sandboxed inference), auditable logs, and human override with accountable final decisions.

"*No central intervention is needed; let selection happen after publication (or via volume controls).*" This view believes the crisis is overstated or self-correcting: low-quality work will be filtered by attention, citations, and replication, and aggressive gatekeeping risks harming open science and disadvantaging under-resourced researchers. *Our response:* this is a valid view, but they do not substitute for scalable verification bandwidth; without evidence-grounded checks, they primarily shift the bottleneck downstream. For example, a minimal, verification-based CRA layer can reduce unnecessary gatekeeping by making validity checks cheaper, faster, and more uniformly available, while preserving high-variance cases for deeper human deliberation.

## 5. CRA in Practice: Lessons from CSPaper Review and the Capability Frontier

CSPAPER REVIEW (CSPR)[1] is an attempt to operationalize CRA in the wild for computer science (CS) venues. CSPR offers free, conference-aware, AI-powered pre-review service that accepts arXiv IDs or PDFs and returns, in under a minute, a venue-specific report comprising: (i) a *desk-rejection assessment*, (ii) an *expected review outcome*, and (iii) *rubric-aligned critical ratings*. CSPR implements a streamlined version of the human-AI co-review engine: validated LaTeX/PDF processors extract structured content; conference/track managers inject official rubrics, calls-for-papers, and benchmarking samples (via a dedicated portal, Figure 4 in Appendix) into agent design; pre-review gate-

---

[1]https://cspaper.org

keepers validate length, scope, and prompt-manipulation risk; and a small committee of review agents is forced to justify divergent ratings (*best-justified*, *more optimistic*, *more critical*) before a calibrated synthesis. CSPR reports that in its first six months online, it served **over 30,000 unique users** from **128 countries** and processed **over 100,000 review requests**, indicating unmet demand for fast, rubric-grounded feedback in CS.

CS is an ideal initial testbed for CRA: (a) conferences dominate dissemination and operate on tight cycles, making *early, actionable* feedback disproportionately valuable; (b) venues publish *standardized review rubrics*, enabling faithful alignment of AI outputs to human expectations; and (c) the community is unusually *open and decentralized* (preprints, artifacts, code), which is conducive to the Community Commons and rapid iteration. While our framework is method-agnostic and not limited to LLMs, CSPR provides an opportunity to study CRA at scale today.

### 5.1. What CRA Can Do Today: Evidence from CSPR

CSPR provides a proof-of-concept that CRA has already delivered practical value at scale to individual researchers. Within a minute, a venue-aware agentic pipeline produces conference-specific triage (desk-reject vs. send-to-review), expected outcomes, and rubric-aligned ratings, surface-level rigor checks (length/scope fit, missing sections/artifacts), highlighting clarity and comparison gaps, and preserving informative variance by synthesizing multiple justified perspectives rather than averaging them away. Reported usage patterns, including repeated submissions of evolving drafts and cross-track probes, indicate latent demand and show CRA functioning as an author-side coach by off-loading routine checks, aligning manuscripts with venue norms, and directing scarce human attention to the most consequential issues, without attempting to replace expert judgment. These priorities align with early CSPR user demand signals (Figure 1) from the first four months after launch.

- *Rapid and venue-aligned triage.* CSPR reliably maps manuscripts to conference-specific rubrics and returns structured feedback in a minute, supporting authors' venue selection and "revise-and-resubmit" planning. For many submissions, this *early triage* replaces weeks of uncertainty with concrete, rubric-grounded next steps.

- *Consistent coverage of surface-level rigor.* Automated checks for length, scope fit, and missing sections or artifacts, together with rubric-aligned ratings, ensure uniform attention to clarity, organization, dataset disclosure, and baseline completeness. This raises the quality floor by catching routine issues that human reviewers often miss under time pressure.

- *Broad literature and context cues.* Even without perfect novelty judgment, CSPR surfaces up to ten directly re-

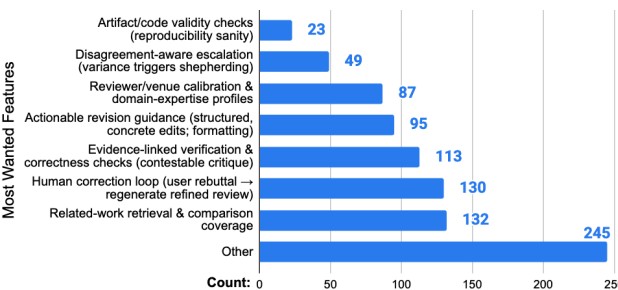

*Figure 1.* Most-wanted features from CSPR user feedback (874 informative, valid responses in total), collected via a survey form and proactive outreach emails. Free-text responses were manually mapped to a fixed set of feature categories.

lated works and venue-specific norms for the selected track, guiding authors toward missing comparisons and more appropriate positioning.

- *Constructive, divergent perspectives.* By forcing agents to justify *different* plausible ratings and then calibrating them (Cao et al., 2025), CSPR exposes variance in evaluation rather than averaging it away, highlighting potential "high-risk/high-reward" submissions that deserve closer human attention.

- *Observable author iteration loops.* Usage patterns show repeated reviews of evolving drafts and the same paper across multiple tracks. This supports CRA's role as an *author-side coach*: fast feedback drives measurable improvements *before* formal submission.

### 5.2. Near-Term Extensions

Given CSPR's core architecture introduced in (Cao et al., 2025), several capability upgrades are within near reach through incremental engineering and light human feedback: attaching evidence links and simple verifications (e.g., citation consistency, statistical sanity checks, image forensics, and code validity) to make critiques immediately contestable; turning multi-agent variance into an explicit disagreement signal that triggers shepherding rather than silent averaging; adding transparent, venue-aware novelty indicators that combine semantic and citation-graph features; calibrating models with lightweight reviewer/author ratings and override justifications; and using a Community Commons to benchmark, red-team, and continuously improve its robustness. These extensions aim to keep CRA assistive and auditable while expanding its reliability envelope.

- *Evidence-grounded verification and correctness checks.* Augment critiques with explicit evidence links by integrating citation-consistency checks, statistical sanity tests, and logic-based claim verification to assess whether key conclusions are supported by the paper content, making AI feedback falsifiable and contestable.

- *Artifact and code validity checks.* Automatically assess submitted code artifacts for basic validity, relevance,

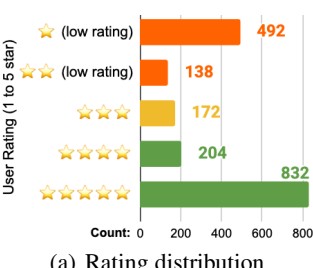
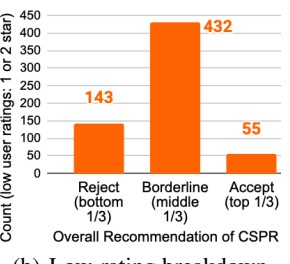

(a) Rating distribution     (b) Low-rating breakdown

*Figure 2.* Post-review quality ratings for CSPR outputs collected after delivery, used as a lightweight outcome signal for revision-utility and systematic auditing of low-rated cases.

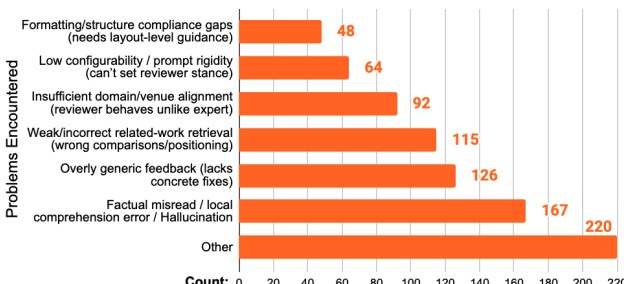

*Figure 3.* Problems encountered in CSPR user feedback (832 informative and valid responses for this question), collected via a survey form and proactive outreach emails. We manually mapped free-text responses to a fixed set of issue categories.

and completeness relative to the paper, flagging placeholder implementations, mismatches, or missing components that may hinder reproducibility (Baumgärtner & Gurevych, 2026).

- *Disagreement-aware escalation.* Generalize multi-view agents into an explicit *disagreement signal* (variance across agents and humans) that triggers shepherding or additional expert review rather than silent averaging.

- *Transparent, venue-aware novelty indicators.* Combine semantic embeddings with citation-graph features (e.g., atypical concept pairings and domain-bridging distances) to surface interpretable novelty cues that are tunable to venue norms rather than opaque scores.

- *Human-in-the-loop calibration.* Add lightweight support for reviewer/author feedback (post-review quality ratings; override justifications; Figure 2) and a post-review correction step that lets users flag factual misreads or missing context, after which the system regenerates a refined review; use these signals to calibrate by subfield and learn where humans systematically add value.

- *Commons-driven robustness and oversight.* Leverage the Community Commons to host red-teaming cases, benchmarks, and evolving policy, enabling continuous monitoring and improvement of robustness to errors, bias, and adversarial manipulation under community governance.

The hardest fronts lie beyond incremental tooling: discerning deep novelty and long-horizon impact outside training distributions; ensuring contestability and reliability when automated assessors can be brittle or factually wrong; mitigating subfield/style bias and drift under rapid domain change; resisting adversarial manipulation and metric gaming (including prompt-injection-style attacks on automated reviewers); governing sensitive, unpublished materials with rigorous privacy and access controls; evaluating CRA itself when ground truth is noisy, delayed, or normative; and adapting beyond CS to fields with different evidentiary standards and weaker rubric traditions. Recent overviews of AI peer review (Shah, 2022) and CSPR user feedback (Figure 3) highlight both long-standing biases and concrete failure

modes of automated reviewing, reinforcing that these are system-design problems, not just "bigger model" problems.

### 5.3. Hard Problems and Open Risks

- *Deep novelty and long-horizon impact.* Recognizing paradigm-shifting ideas that deviate from training distributions remains intrinsically hard; worse, automated reviewers may latch onto spurious surface cues rather than scientific substance. CRA should therefore *surface* high-upside candidates via disagreement and variance, rather than claim autonomous adjudication.

- *Hallucinations, reliability, and contestability.* Any automated critique can be faulty (Dycke & Gurevych, 2025). Empirically, AI-review pipelines can struggle with factual accuracy unless prompted at the right granularity, and they may fail to reject even obviously nonsensical "chimera" manuscripts (Shah, 2024). As pointed out by (Naddaf, 2025a), CRA outputs must be *contestable by design*: traceable evidence, uncertainty signals, and easy human override.

- *Ground truth and evaluation.* Robustly measuring review quality is non-trivial: acceptance is an imperfect label, human reviews are noisy, and impact is delayed. Moreover, even seemingly principled aggregation of multiple criteria can behave pathologically, such as "commensuration" issues (Shah, 2024), indicating the need of venue- and subfield-specific evaluation protocols (e.g., revision-utility, error-catch rate, human-AI complementarity) reported transparently (see CSPR post-review quality ratings in Figure 2).

- *Generalizing beyond CS.* Extending CRA beyond computer science requires "adapting to fields with different publication cultures" (Adam, 2025) by defining domain-specific dimensions (He, 2024), reweighting criteria, and engaging new communities through the Commons; it also requires careful attention to identity- and prestige-linked biases and to experimental design that can detect them under realistic constraints.

### 5.4. What a CRA System Should Not Do

A mature CRA system acts as a guardian of quality and efficiency, handling the tedious and highly structured aspects of review with superhuman consistency. Yet, there are few things that we believe should not be part of a CRA system.

- *Replace human creativity and judgment*. As we have emphasized, CRA will not supplant the human element in recognizing groundbreaking science. It cannot *define the very frontier of knowledge* – humans (scientists themselves) push that frontier, and humans will be needed to recognize when something truly novel and important has arrived. If an AI could *perfectly* judge the long-term impact of a discovery, it would have to be as intelligent and creative as the entire field's community (Bergstrom & Bak-Coleman, 2025). That day is far off, if it ever comes (Lindsay, 2023). Until then, we explicitly keep humans as the final arbiters, especially when it comes to that gut feeling of "this idea is a game-changer". The CRA's purpose is to assist and inform, not to assert its verdict over the objections of experienced researchers.

- *Be completely objective or value-free*. One might hope that introducing computation removes subjectivity, but in truth, values and subjective choices will simply be encoded into the system. For instance, how do we weight novelty vs. technical rigor? That's a subjective policy decision. CRA cannot tell us what we *should* value; the scholarly community must decide that and bake it into the algorithms. CRA systems also cannot escape the biases of their training data – if most past reviewers undervalued a certain type of research (say qualitative studies or negative results), the AI might learn that same bias (Aczel et al., 2025). We can mitigate this, but not perfectly. There will always need to be human oversight to examine biases that creep into the AI's outputs.

- *Guarantee correctness of its own outputs*. Paradoxically, an AI can analyze a paper and still be wrong in its analysis. We see this with LLMs that produce plausible but incorrect critiques. CRA systems will likely produce *false positives* (flagging an issue that isn't actually an issue) and *false negatives* (missing a subtle problem). For example, an AI might misunderstand a novel methodology and flag it as an error simply because it's unusual. Or it might miss a complex logical flaw that requires domain insight. Therefore, CRA outputs will *always require human validation*. The goal is to reduce human load, not to eliminate human thinking. In practice, we should treat AI flags as suggestions or hypotheses for the human reviewer to verify, not as absolute truths.

- *Fully eliminate strategic behavior*. We can strive for immunity to gaming, but some gaming will always exist. Researchers will inevitably try to optimize how their papers score on known criteria. In fact, *some of that is positive* – if authors write more clearly and do more thorough literature reviews because they know the AI will check, that's a *win* for everyone. But other strategic behavior might be neutral or negative (e.g., overselling the novelty with grandiose language because they think the AI counts adjectives, or adding unnecessary citations of popular papers to show connectivity). The CRA immune system can catch blatant ploys (like hidden text or gibberish insertion to fool metrics), but the subtler ones may slip through, at least initially. This is similar to cybersecurity: there will be a continual arms race between new attack (gaming) techniques and defenses. We have to accept that CRA won't end the game of "please the reviewer/AI" – it just changes its form. With vigilance via the Commons, we hope to minimize perverse incentives, but we shouldn't be naive that we can avoid them all.

- *Make unpopular decisions without backlash*. If an AI-influenced process rejects a paper that a segment of the community really believes in, there could be pushback. People might distrust the AI's role ("my paper was misunderstood by a machine!"). Even if the decision was ultimately human, the AI's fingerprints on it provide an easy target for authors to blame. Thus another thing CRA cannot do is avoid controversy. In fact, as it surfaces more issues and perhaps recommends bolder acceptance of risky ideas, it might stir *more* debate in the short term. This is not necessarily bad since open debate is healthy; but we should be prepared that CRA will have to earn trust over time. It cannot demand trust on day one; it has to earn the credibility that human peer review, for all its shortcomings, has built over centuries.

These limits keep the proposal grounded: CRA is a toolset to strengthen human-led evaluation, not an oracle. Humans remain the accountable decision-makers and value-setters, and CRA must be designed for override, opt-out, and continuous monitoring in deployment. This motivates our concluding thesis: peer review should treat disagreement as a signal, and tie every critique to contestable evidence.

## 6. Conclusion

Peer review, as a social process, needs help at scale. Artificial intelligence can provide breadth, speed, and consistency. Humans provide creativity and responsibility. The right question is not whether machines can replace reviewers. The right question is how to design a process where disagreement becomes a resource and where every claim is tied to evidence. Computational Research Assessment (CRA) answers this question with a co-review engine, a community commons, and a formal core. The program is small enough to pilot now. It is broad enough to guide the next decade. We invite the community to test it, to attack it, and to improve it.

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

## A. Venue Management Portal (VMP)

In CSPR, conference/track managers inject official rubrics and call-for-papers via a dedicated venue management portal (VMP) that allows organizers to create, iterate, and maintain venue-specific review agents (Figure 4). Through the VMP, managers can edit venue meta info, allowed topics, reviewer profiles, score calibration, expected paper structure/composition, and the review template itself (including adding/reordering/removing sections, per-section instructions, and rating options).

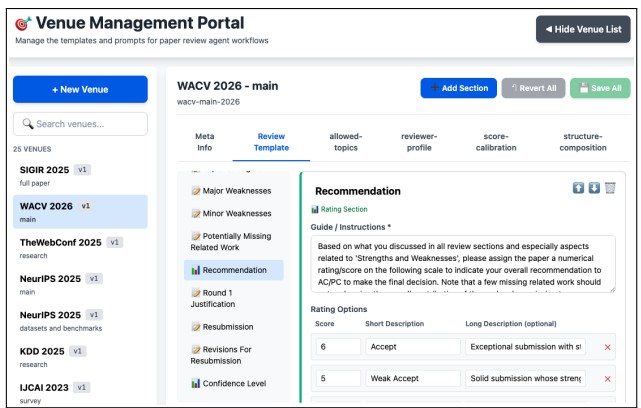

*Figure 4.* The Venue Management Portal (VMP) used by conference and track managers to configure, iterate, and maintain venue-specific review agents, including meta information, reviewer profile, review template/guideline, compatible topics, paper composition, and calibration settings.

