# OpenReview forum: "Position: Adopt Machine-Human Collaboration Peer-Review through Computational Research Assessment"
_ICML.cc/2026/Position_Paper_Track — Submitted to ICML 2026 Position Paper Track_

### Official Review · Reviewer_Y2S6 · 2026-03-05

**Significance:** 3
**Argument Clarity:** 3
**Rating:** 5
**Confidence:** 4

**Questions:**

Please see above - I left several comments for the authors to address.

**Alternative Views Section:**

Yes

**Compliance With Llm Reviewing Policy A Conservative:**

Affirmed.

**Discussion Potential:**

3

**Final Justification:**

I don't have more comments!

**Paper Summary:**

This was a well-written and thought-provoking paper on human-AI teams for reviewing papers - computational research assessment (CRA). It discusses interesting ideas like quantifying rigor, novelty and impact and recommends handling disagreements between the human and AI more carefully than just averaging.

**Position:**

Yes

**Position In Title:**

Yes

**Related Work:**

3

**Strengths And Weaknesses:**

I think this paper should be accepted, however, I do have some important comments that I would like the authors to address.

1) Reviewer appropriateness:
First, the author(s) seem to assume that the reviewers are competent and well-meaning but just time-constrained. I am absolutely certain that this is not generally the case!
The author(s) write: "We risk missing breakthrough ideas or letting flawed work slip through because overwhelmed reviewers cannot cope." <- it's not that they can't cope. It's that they don't care. The community has turned into something that simply just doesn't care about peer review any more. Once too many sloppy and inexperienced reviewers entered the system, it's hard to manage the process. It's a serious cultural problem! Also the reviewer matching is really really bad, so papers in slightly uncommon fields get reviewers who are not familiar with the literature - those reviewers are worse than AI because they just reject the paper because they can't understand it. And there are tons of unqualified reviewers. Those reviewers will simply let the AI write the review for them, so that the authors are just responding to AI. Let's say the reviewer outsources the review to AI (which currently happens a *lot*). Then the author feels like they are just arguing with a machine, which is an exercise in frustration. Hence ICML's option 1 for non-LLM reviewing.

One solution to this is to have the LLM help determine from the reviewer's publication record whether they have the expertise to review this particular paper. Another is to ask the LLM to determine whether the reviewer actually contributed before the review is submitted.

LLMs should also flag whether reviews could be malicious - someone reviewing to make sure their competitor doesn't publish first.

2) Automation bias:
This is related to my previous point. How do you know that the AI won't just cause the reviewers to be really sloppy? They are already sloppy. It is easier to outsource to AI. So why wouldn't they? You write that the human is accountable. Since when does that matter in reality? If the human hands in a review that looks ok on the surface, who will hold them "accountable" and what does that mean?

3) Quantifying novelty: The idea of quantifying novelty and impact is really cool but I have no idea how that can be done reliably. Certainly reviewers should reference other papers when providing evidence that a paper is not novel.

4) Tail papers:
Papers in the "long tail" already have a hard time, and I suspect using AI for reviewing would make it worse. The AI wouldn't know how to handle those papers, and neither do the reviewers. Those papers would thus have an even more difficult time compared to other papers.

5) Copyright:
Is there any serious concern about copyright? It was mentioned only in passing. I don't think it's ok to send someone's unpublished paper to a third party LLM without permission, and I don't think it's ok for that LLM to train on that paper. In fact, I think those are truly awful. How do we know that won't happen? Do we trust those companies, given what they've already done with copyrighted material, against the authors' wishes?

**Support:**

3

---

> ### Author Rebuttal · Authors · 2026-03-27
>
> Thank you for the supportive review and for the particularly concrete comments. We agree these issues should be stated more strongly in the paper.
>
> Reviewer quality / culture: we agree the problem is not only reviewer bandwidth. Reviewer mismatch, weak engagement, and even malicious reviewing are part of the setting. We will enhance this point to say this explicitly. CRA should therefore include safeguards around reviewer expertise profiling, mismatch detection, contribution/accountability checks, and possible malicious-review flags (with human oversight), rather than assuming ideal human participants.
>
> Automation bias / reviewers outsourcing to AI: agreed. Our position is not that "human in the loop" alone solves this. We will strengthen the text to require bias-aware deployment: evidence-first AI outputs, optional hiding of scalar AI scores until after an initial human pass, required override justifications, and escalation when human/AI or reviewer/reviewer disagreement is large. The purpose is to make blind AI outsourcing harder, not easier.
>
> Long-tail papers: we share this concern. In fact, these are exactly the cases where CRA should reduce automation and increase escalation. We will add that atypical or low-confidence papers should trigger more human attention and specialized reviewer recruitment, not stronger automated confidence.
>
> Novelty / impact: we agree reliable quantification is hard. Our claim is narrower: CRA should surface interpretable novelty evidence (nearest prior art, unusual combinations, domain bridges) and uncertainty, not pretend to solve long-horizon impact prediction.
>
> Copyright / confidentiality: we strongly agree and will make this explicit. For unpublished manuscripts, venue-side CRA must use strict data controls (e.g., on-prem/sandboxed inference or author-consented processing) and must not send papers to third-party training endpoints without permission. CSPR is author-side and opt-in; venue-side deployment requires a stricter governance bar.
>
> Your suggestions on reviewer expertise matching and malicious-review flags are excellent, and we will incorporate them into the governance/open-risks discussion. Thank you again for the positive assessment.

---

> > ### Author Rebuttal · Reviewer_Y2S6 · 2026-04-01
> >
> > I hope my comments can be put into any revision of the paper because I think they are important.

---

### Official Review · Reviewer_wpE7 · 2026-03-10

**Significance:** 2
**Argument Clarity:** 2
**Rating:** 4
**Confidence:** 2

**Questions:**

See weakness

**Alternative Views Section:**

Yes

**Compliance With Llm Reviewing Policy A Conservative:**

Affirmed.

**Discussion Potential:**

3

**Final Justification:**

after discussion, some of my issue is resolved

**Paper Summary:**

This position paper argues that the peer-review system in machine learning is facing a scalability crisis due to rapidly increasing submission volumes and limited reviewer capacity. To address this issue, the authors advocate for Computational Research Assessment (CRA), a framework that promotes machine–human collaboration in the peer-review process.

The proposed framework consists of three main components: (1) a human–AI co-review engine, where AI systems perform structured pre-review analysis to assist human reviewers; (2) a community commons for datasets, benchmarks, and red-teaming to support the development of AI-assisted review tools; and (3) theoretical foundations for computationally evaluating research quality, including aspects such as novelty, rigor, and impact.

The paper outlines a review workflow in which AI systems provide preliminary assessments and human reviewers interact with and refine these evaluations. Overall, the authors position CRA as a discipline-level agenda aimed at improving the scalability and effectiveness of peer review through structured collaboration between AI systems and human reviewers.

**Position:**

Yes

**Position In Title:**

Yes

**Related Work:**

3

**Strengths And Weaknesses:**

Strength:

1. The paper proposes the peer-review scalability crisis in machine learning: (1) Submission numbers to major conferences are rapidly increasing.  (2) Reviewer capacity does not scale at the same rate.

2. The paper explicitly considers opposing perspectives, including:  AI should only perform verification, AI should not participate in peer review, and Post-publication filtering may suffice.

Weakness:

1. While this paper argues that using AI review to reduce the human workload, the efficiency gains are unclear. Specifically, The pipeline design assumes: AI review → human review → better efficiency. In lines 152-153, "Conversely, if the AI missed something, say a
subtle logical gap, the reviewer can add that to the report." This still requires a human to carefully read and verify the paper to find the missing part of the AI review.  I believe more discussion is warranted.

2. The core ideas of the paper overlap with existing research on AI-assisted peer review, like [1], so it is not a new position but an already actively studied area.

[1] ReviewerToo: Should AI Join The Program Committee? A Look At The Future of Peer Review

**Support:**

2

---

> ### Author Rebuttal · Authors · 2026-03-27
>
> Thank you for the thoughtful review and for recognizing both the scalability problem and the value of engaging alternative views.
>
> Your main concern is efficiency. We agree the paper should be more precise: the claim is not "AI reviews, therefore humans can stop reading". The claim is that structured, evidence-linked pre-review can reduce blank-page effort and standardize routine checks (scope/format fit, missing artifacts/sections, related-work surfacing, citation consistency), so human attention is spent on interpretation, novelty judgment, and disagreement resolution. For author-side pre-review, the largest efficiency gains occur before formal submission; for venue-side use, the gain is in concentrating scarce human attention on high-uncertainty cases. We will make this distinction explicit and add concrete evaluation criteria such as reviewer time per paper, turnaround, revision utility, and error-catch rate.
>
> On novelty relative to prior AI-assisted peer-review work such as ReviewerToo: we agree the related-work comparison should be sharper and will add it. We do not claim to invent the area of AI-assisted reviewing. The novelty of this position paper is the broader CRA framing: (i) a method-agnostic, discipline-level agenda rather than a single LLM tool; (ii) disagreement as a signal that should trigger escalation rather than be averaged away; (iii) evidence-linked, contestable critiques; and (iv) a community commons plus theoretical foundations for benchmarking, governance, and robustness. We will explicitly contrast this with prior work that studies whether AI should join the PC or assist reviewing, and cite ReviewerToo directly.
>
> Thank you again; if these clarifications resolve your main concerns, we hope you would consider moving the score upward.

---

> > ### Author Rebuttal · Reviewer_wpE7 · 2026-04-03
> >
> > Thank you for your response. I believe these points are quite essential, as the author acknowledge limitations in efficiency, which weakens the paper’s overall position, so I keep my original score

---

### Official Review · Reviewer_eghd · 2026-03-15

**Significance:** 3
**Argument Clarity:** 2
**Rating:** 1
**Confidence:** 4

**Questions:**

Please see weaknesses. Thank you.

**Alternative Views Section:**

Yes

**Compliance With Llm Reviewing Policy A Conservative:**

Affirmed.

**Discussion Potential:**

2

**Final Justification:**

Many of the clarifications provided in the rebuttal would benefit from being more explicitly and precisely incorporated into the paper itself.

**Paper Summary:**

The paper advocates for machine-human collaboration for peer reviewing, and suggests a new system called Computational Research Assessment.

**Position:**

Yes

**Position In Title:**

Yes

**Related Work:**

2

**Strengths And Weaknesses:**

Strengths:
- I appreciate that the paper tries to address problems in current paper reviewing systems.

Weaknesses:
- In the abstract, what does it mean to have a “production-grade pre-review system”?
- (nitpick) in line 16, “averaging review scores is no longer defensible …”, I don’t think the practice of averaging review scores was ever defensible. A lot of convenient practices are done in practice, but not defensible when there are more resources.
- Line 23, what is a “discipline-level program”? What does “discipline-level” mean?
- “The aim of peer review … should not be to select the least objectionable papers, but to identify outstanding contributions even when they are polarizing and imperfect”. I agree that peer review should identify outstanding contributions even when they are polarizing and imperfect, however, the peer review system should also select the least objectionable papers. Those papers are still good science. Hence, the phrasing can be made better with “not only…. but also …”
- While the paper sets up the general limitations of the current peer review system, it is not clear how we should measure how well a peer review system is doing, and hence, It is not clear why the proposed system CRA is better than the current system and how we would even know that.
- Line 109, "specialized AI agents configured for the venue and field”. The paper can discuss more about why the venue matters? and should it? Just like how the paper has already pointed out that peer review is heavily influenced by an individual’s biases, doesn’t the venue matter not because of any objective measure, but because the people who typically review papers (e.g., reviewers, ACs, PCs) have certain biases, which is then reflected in the venue? The AI conference venues typically accept paper submissions from similar fields and contributions, even though the accepted papers are often of a different flavour, reflecting the subjectiveness in each venue.
- The biggest question I have for this position paper is how would the authors know that their proposed system is better than the current peer review system? If it is by measures chosen by the authors, isn't it unfair? whereby the measures should first be accepted widely as a scientific community before accepting that another system is better
- Line 158, “the drudge work is largely done”, referring to AI checking for similar papers and reevaluating novelty as doing the drudge work. Is this really where reviewers spend most of their time and consider it as drudge work? As a reviewer myself, I consider the most drudge work as rewriting my thoughts into sentences that would make sense for the authors (e.g., making sure that my feedback is actionable and constructive, phrased in a kind way to still acknowledge the amount of work that they have put in).
- Line 117, “prompt a second human opinion or a moderated discussion”. I can foresee that one side effect is that reviewers would then more likely want to agree with the AI reviewer, given how busy reviewers are.
- line 185 about community commons (CC). Wouldn’t the CC inevitably be games too? How do we know if the CC helped the peer review system or is a suitable CC to follow?
- line 266, “start to treat peer review itself as a subject of computational and theoretical study rather than just an art”. Earlier, the paper suggests that scientific value is subjective, and hence we need diverse evaluation (section 2). However, now it seems like the paper is trying to advocate for treating peer review as a subject of computational and theoretical study.The paper is contradicting itself.

**Support:**

2

---

> ### Author Rebuttal · Authors · 2026-03-27
>
> Thank you for the careful read. Several of your comments are helpful and, in our view, point to places where our wording was too loose. We can fix that.
>
> To be direct on the main point: the paper is **not** claiming a finished system that already outperforms peer review. It is a position paper proposing a research agenda and explicit evaluation/governance criteria because ad hoc AI-assisted reviewing is already happening.
>
> ### 1. "How would we know CRA is better? If the authors choose the measures, isn't that unfair?"
>
> We agree. Because this is a position paper, our claim is not that CRA is already proven superior, nor that authors should invent a bespoke metric and declare victory. Our claim is that CRA should be judged by community-accepted protocols. The paper already points toward these (revision utility, error-catch rate, human-AI complementarity, reviewer time per paper, override/calibration behavior, and downstream corrections/retractions), but we agree this needs to appear much earlier and more sharply. This is also exactly why the Community Commons is part of the proposal: open, community-governed benchmarks and policy deliberation should come before strong claims of superiority.
>
> ### 2. "discipline-level program" / "production-grade pre-review system"
>
> By "discipline-level" we mean an agenda about how a scientific discipline evaluates papers - not a tool for one venue, one model family, or one company. By "production-grade pre-review system", we mean a publicly deployed system used by real authors at scale, rather than a lab prototype. We will define both terms explicitly in the abstract/introduction.
>
> ### 3. "Averaging review scores is no longer defensible" / "not least objectionable but outstanding…"
>
> Fair point on phrasing. We will revise to: peer review should not **only** select the least objectionable papers; it must **also** preserve a path for high-variance, high-upside work. Likewise, "no longer defensible" may be too absolute; the intended point is that simple averaging is especially inadequate under high variance and rising scale.
>
> ### 4. Why venue-aware? Doesn't that just encode venue bias?"
>
> Venue-awareness, as we intend it, should track explicit published rubrics and expectations (scope, artifact requirements, evaluation criteria), not sanctify implicit bias. In fact, making venue assumptions explicit and auditable is preferable to leaving them hidden in informal practice. We will clarify that venue conditioning must be limited to explicit criteria and remain overrideable; it is not "the venue is always right".
>
> ### 5. The paper is contradictory: value is subjective, yet you want computational/theoretical study."
>
> We respectfully **disagree** that these are contradictory. Recognizing that scientific value is partly subjective is exactly why **CRA should model disagreement, uncertainty, and plurality rather than pretend there is one objective scalar truth**. Theoretical study here means formalizing what can be formalized, identifying what should remain contestable, and studying aggregation/escalation under disagreement.
>
> ### 6. "drudge work" / automation bias / commons can be gamed
>
> We agree that "drudge work" was too narrow and that human writing labor should be acknowledged. We will revise this to "routine structured checks". We also agree that automation bias and gaming of the Commons are real risks; the paper's intent is precisely that these be first-class design problems. We will strengthen the text on hiding scalar AI scores before an initial human pass, requiring overrides/justifications, and using the Commons for red-teaming and audit rather than treating it as inherently trustworthy.
>
> Thank you again. We appreciate the push for sharper definitions and evaluation criteria, and if these clarifications address your concerns, we hope you would consider revising the score.

---

> > ### Author Rebuttal · Reviewer_eghd · 2026-04-04
> >
> > please see my comment, thank you

---

### Official Review · Reviewer_dSSL · 2026-03-16

**Significance:** 4
**Argument Clarity:** 3
**Rating:** 5
**Confidence:** 3

**Questions:**

While the discussion on the conflicts between polarization in opinion vs consensus in AI generation in scientific discovery is valid, how would other factors, such as the quantity, speed and imbalance use of (AI) resource in different scientific fields affect the proposed paradigm?

Does the proposed framework introduce proper leverage that can adjust the balance between human and AI involvement in each of the components?

**Alternative Views Section:**

Yes

**Compliance With Llm Reviewing Policy A Conservative:**

Affirmed.

**Discussion Potential:**

3

**Paper Summary:**

This paper advocates a new paradigm Machine-Human collaboration peer-review via Computational Research Assessment (CRA). The authors identifies the challenge of reviewing large quantity of scientific discovery with AI augmentation and assistance. The authors proposed CRA as a discipline-level method agnostic agenda for human-AI collaboration in peer-review, with three pillars: human-AI co-review engine, a community commons, and theoretical foundations.

**Position:**

Yes

**Position In Title:**

Yes

**Related Work:**

3

**Strengths And Weaknesses:**

Strengths:
This paper studies an important problem and emerging urgency of review and assessment scientific discovery with AI-human collaboration.

The paper is well structured and has in-depth discussion on why it is needed.

The paper provides realistic approaches and framework that outlines future directions.

Weaknesses:
It would be good to see more in-depth discussion on why Human-AI co-assessment on scientific discovery is needed. While the discussion on the conflicts between polarization in opinion vs consensus in AI generation in scientific discovery is valid, how would other factors, such as the quantity, speed and imbalance use of (AI) resource in different scientific fields affect the proposed paradigm?

It would be good to see more in-depth discussions on how different scientific fields and practices, e.g., physics vs computer science, empirical vs theoretical discoveries affect the co-assessment approach. Does the proposed framework introduce proper leverage that can adjust the balance between human and AI involvement in each of the components?

The introduction of stage-A where AI agents generate rapid pre-review has a strong assumption, where human assessment can be unbiased towards AI generated pre-review. There is likely propensity of human assessment towards AI assessment. We need either bias aware AI pre-review or human pre-review without AI input to adjust for such propensity.

While one of the most important aspects of the proposed position is to accelerate review speed with effectiveness due to the overwhelming growth of AI generated content. It would be good to see some discussion on how the proposed CRA framework can save overall assessment time, compared to current human-only review, or current unstructured human review with AI input.

**Support:**

3

---

> ### Author Rebuttal · Authors · 2026-03-27
>
> Thank you for the positive assessment and for highlighting the paper‘s importance, structure, and realism. We are glad the core position landed.
>
> Your central concern is whether CRA gives enough leverage to vary human vs. AI roles across fields, and whether Stage A risks automation bias. Yes - CRA is intended to be adaptive, not fixed. We will make this explicit. The AI/human split should depend on rubric strength, artifact openness, field maturity, confidentiality, and uncertainty. In rubric-heavy CS venues, AI can do more structured pre-checks; in theoretical or long-tail settings it should be limited to bounded verification/literature surfacing while humans dominate judgment. Those levers may present through venue-specific rubrics, reviewer profiles, score calibration, allowed-topic constraints, escalation thresholds, etc.; we will foreground that these are exactly the knobs that adjust human vs. AI involvement.
>
> On automation bias: we agree this needs to be stated more strongly. We will revise Stage B to say that AI pre-review should be evidence-first, not verdict-first; venues may hide scalar AI scores until after an initial human pass; reviewers must justify agreement/override; and strong human-AI disagreement should trigger escalation rather than averaging. In other words, the system should be bias-aware by design.
>
> On efficiency: our claim is not that AI removes the need for careful reading. The gain is that routine checks (format/scope fit, missing sections/artifacts, related-work surfacing, citation consistency, code validity) can be done early and uniformly, so human effort is concentrated on interpretation and judgment. We will state concrete evaluation criteria: reviewer time per paper, turnaround, revision utility, and error-catch rate, rather than vague “faster review.”
>
> We will also add a short paragraph on resource imbalance across fields: CRA should not assume equal AI resources everywhere. This is precisely why we frame CRA as a discipline-level agenda with adjustable deployment modes, including minimal verification-first setups for fields with weaker rubrics or tighter privacy constraints.
>
> Thank you again; these enhancements will strengthen the paper further.

---

> > ### Author Rebuttal · Reviewer_dSSL · 2026-04-05
> >
> > Thank you for the rebuttal! The paper studies an very important matter while improvement could be made for further enhancing the points made in this paper.

---

### Decision · Program_Chairs · 2026-04-30

**Decision:**

Reject

**Comment:**

The paper had an extreme range of reviewer feedback, 3 positive and one negative.
One reviewer stressed the need for more in-depth discussion about the many issues, but accepted anyway.  Another reviewer mentioned "ReviewerToo: Should AI Join The Program Committee? A Look At The Future of Peer Review", a high profile study of ICLR data.  This raises the issue of machine-human collaboration and gives a set of suggestions backed by ICLR data.  Missing this paper is a serious problem for the paper.
I believe the proposal is a bit too specific, and alternatives for human-AI collaboration are needed.
Nonetheless, the paper generated a lot of valuable discussion and comments from the reviewers.